# Codon Usage Analyses Reveal the Evolutionary Patterns among Plastid Genes of Saxifragales at a Larger-Sampling Scale

**DOI:** 10.3390/genes14030694

**Published:** 2023-03-11

**Authors:** De Bi, Shiyun Han, Jun Zhou, Maojin Zhao, Sijia Zhang, Xianzhao Kan

**Affiliations:** 1Suzhou Polytechnic Institute of Agriculture, Suzhou 215000, China; 2Anhui Provincial Key Laboratory of the Conservation and Exploitation of Biological Resources, College of Life Sciences, Anhui Normal University, Wuhu 241000, China; 3The Institute of Bioinformatics, College of Life Sciences, Anhui Normal University, Wuhu 241000, China

**Keywords:** Saxifragales, plastid genes, phylogeny, codon usage

## Abstract

Saxifragales is a 15-family order of early-divergent Eudicots with a rich morphological diversity and an ancient rapid radiation. Codon usage bias (CUB) analyses have emerged as an essential tool for understanding the evolutionary dynamics in genes. Thus far, the codon utilization patterns had only been reported in four separate genera within Saxifragales. This study provides a comprehensive assessment of the codon manipulation based on 50 plastid genes, covering 11 constituent families at a larger sampling scale. Our results first showed a high preference for AT bases and AT-ending codons. We then used effective number of codons (ENC) to assess a range of codon bias levels in the plastid genes. We also detected high-informative intrafamilial differences of ENC in three families. Subsequently, parity rule 2 (PR2) plot analyses revealed both family-unique and order-shared bias patterns. Most importantly, the ENC plots and neutrality analyses collectively supported the dominant roles of selection in the CUB of Saxifragales plastid genes. Notably, the phylogenetic affinities inferred by both ML and BI methods were consistent with each other, and they all comprised two primary clades and four subclades. These findings significantly enhance our understanding of the evolutionary processes of the Saxifrage order, and could potentially inspire more CUB analyses at higher taxonomic levels.

## 1. Introduction

The order Saxifragales is the representative of parts of the early-divergent branches of the Eudicot [1,2,3]. Fossil data have suggested a rapid radiation of this group [2,4], and currently, it includes ca. 2500 taxa from 15 families [5], exhibiting a high degree of diversity in morphology (e.g., succulents, shrubs, vines, and trees) [6,7,8]. Although the monophyly of this order has been widely accepted [9,10,11,12,13,14], its internal phylogenetic affinities were more elusive in earlier studies, which might be caused by an ancient, rapid radiation and limited data resources [15,16]. In particular, the circumscriptions of Saxifragales have been regarded as a major surprise in plant systematics [17]. Notably, based on a 50,845 bp concatenated dataset from nuclear, mitochondrial, and plastid sequences, Jian et al. [1] firstly inferred three major clades of Saxifragales: (1) Peridiscaceae, (2) Paeoniaceae + “woody group” (Altingiaceae, Cercidiphyllaceae, Daphniphyllaceae, and Hamamelidaceae), and (3) Crassulaceae alliance (Crassulaceae, Haloragaceae, Penthoraceae, and Tetracarpaeaceae) + Saxifragaceae alliance (Saxifragaceae, Grossulariaceae, Iteaceae, and Pterostemonaceae). Thus far, many analyses have supported this backbone phylogeny, and greatly contributed to insights into the interfamilial relationships among these lineages [18,19,20].

Saxifragales is considered to be a pretty good demonstration to explore diverse evolutionary patterns among angiosperms [17,20]. Along with rapid progress in molecular methods, the evolutionary history of Saxifragales has undergone many intense explorations [18,19,20,21,22,23]. Therefore, plastid genome data exemplifies the advantages of affordability, high sequencing efficiency, and rich evolutionary information. Numerous studies on plastomes have been dedicated to explore genomic organizations, interspecific differences, intracellular gene transfer, RNA editing, genetic engineering, and evolutionary analysis [24,25,26,27,28]. At present, plastomes have been regarded as a powerful tool not only for phylogeny inference, but also for DNA barcoding of plants [24,29,30,31,32,33]. Furthermore, as for plastid genes, the patterns of synonymous codon usage could reflect the driving factors in evolution among different taxa [34,35,36].

Among the plastomes of Saxifragales, under the rapid development of DNA sequencing methodology, there have been several advances in understanding the nucleotide and structural variations in gene content, intron indels, plastomic tRNAs secondary structures, and hypervariable regions, etc. [20,21]. Furthermore, our previous work, involving 208 Saxifragales plastomes, has suggested several family- and lineage-specific markers for this order. However, the evolutionary dynamics of specific plastid genes in Saxifragales have not been systematically investigated yet.

As a ubiquitous phenomenon, codon usage bias (CUB), referring to the unbalanced employment of synonymous codons [37,38,39,40], can provided crucial information for deciphering the evolutionary patterns of genes [41,42,43]. Unravelling such bias could be intriguing, usually due to the comprehensive effects of mutation, selection, and genetic drift [44]. Most significantly, the preference of codon usage is suggested to differ among organisms and even within a certain gene [43,45]. Since the early report of codon catalog usage from Grantham et al. [46], a series of indices have been proposed to quantify CUB, containing effective number of codons (ENC) [47], relative synonymous codon usage (RSCU) [48], parity rule 2 (PR2) plot [49], neutrality plot [50], etc. Within Saxifragales, the patterns of codon usage have been reported for several taxa, including Crassulaceae (*Aeonium*, *Monanthes*, and *Crassula*), and Saxifragaceae (*Saxifraga*) [19,51,52]. Of note, our previous study on *Crassula* revealed a genera-specific distribution of ENC values in *matK* genes [52]. To reach a deeper understanding of Saxifragales plastomes, further explorations on a larger sampling scale will thus be crucial.

The present work aims to comprehensively assess the evolutionary dynamics of Saxifragales plastid genes, with all publicly available sequence data. Employing multiple indices, the goals of this study are to elucidate (1) the various codon bias patterns among plastid genes of Saxifragales, (2) the dominant factors in shaping the diverse CUB patterns, and (3) the phylogenetic affinities within the saxifrage order. Most importantly, it is hoped that this study will broaden and deepen, our understanding of the Saxifragales evolution, as well as the patterns of codon usage at higher taxonomic levels.

## 2. Materials and Methods

### 2.1. Data Retrieval, Reannotation, Alignment, and Compositional Analysis

To obtain the widest sampling of available Saxifragales plastomes, 361 plastomes were acquired from the GenBank of NCBI database (https://www.ncbi.nlm.nih.gov accessed on 20 February 2023), including 24 sequences that were reported in our previous studies. Overall, our sampling involved 258 taxa from 64 genera in 11 families of Saxifragales, which are listed in Appendix A along with their accession numbers and genome sizes. Then, under the principles of (1) including the initial codon, the terminal codon, and no internal stop codons, and (2) being a multiple of three and longer than 300 bp in gene size, a total of 50 protein-coding genes (PCG) were filtrated from each plastome for further analyses. After careful checking and correcting of gene annotations by GeSeq [53], CPGAVAS2 [54], and BLAST [55], coding sequences (CDS) alignment was performed using MAFFT v7.505 [56]. The GC contents of the CDSs at different positions, including GC1, GC2, GC12, GC3, and total GC, were also calculated using DAMBE v7.2.25 [57].

### 2.2. The Calculations of Codon Usage Indices

Concurring with McInerney [58], Roth et al. [59], Wang et al. [60], and others, the termination codons were removed from the CDS matrix for further analyses. Using CodonW v1.4.4 for each gene, we focused on the following CUB parameters: RSCU, ENC, and GC content of synonymous codons at the 3rd position (GC3s) [61].

RSCU represents the proportion of the actual frequency to the excepted frequency for a certain codon, and is routinely applied to detect differences in codon use among genes [44,48]. For this value, equality to, greater than, and less than 1 indicates no, positive, and negative bias, respectively.

ENC values are an excellent estimator for absolute synonymous codon bias and reflect the degree of preferential codons in genes [44,62]. With a range of 20–61, ENC values are inversely related to the extent of CUB [47]. In addition, considerable bias can be observed for values no higher than 35 [63,64].

### 2.3. Graphic Analyses

To visually assess the driving factors of the CUB patterns among plastid genes within Saxifragales, several plot analyses, including a PR2 plot, neutrality plot, and ENC–GC3s plot, were carried out using ggplot2 in R v4.0.1 [65].

The PR2 plot analysis is performed to estimate the influences of mutation and selection on the third position of codons, which is usually confined to the four-fold degenerate codons [42,44,49]. This plot is generated by GC bias [G3/(G3 + C3)] and AU bias [A3/(A3 + U3)] as the *x*- and *y*-axis, respectively [66,67]. In detail, the center-located points suggest no bias and equal effects from both mutation and selection, whereas the off-center splashes would manifest the corresponding orientations and degrees of bias [42,44,68]. More specially, if A3 + T3 is equal to G3 + C3, then the preferential utilization of codons might be entirely influenced by mutational pressure [69].

As a routine analysis to infer the determinants of CUB, ENC–GC3s plot is drawn by GC3s (*x*-axis), ENC values (*y*-axis), and a standard curve (calculated following Wright et al. [47]). In this plot, the splashes will fall near the curve when the CUB patterns are mainly driven by mutational effects. Instead, if the codon preference is determined by selection or other factors, the points will be distant from the curve [70,71,72].

Likewise, a neutrality plot, based on the regression analysis of GC12 against GC3, is also utilized to evaluate the influences from mutation and selection [50]. Furthermore, if the slopes of the regression line are close to 0, selection is suggested to be the decisive force governing the codon usage, while being close to 1 is indicative of the dominating impact of mutation in CUB.

### 2.4. Phylogenetic Inferences of Saxifragales

To reconstruct the evolutionary affinities within Saxifragales, phylogenetic analyses were performed among all 258 involved species by maximum-likelihood (ML) and Bayesian inference (BI) methods. With two taxa from Rosids as outgroups, a sequence matrix of 79 plastid genes from 363 individuals was built using MAFFT v7.505 for multiple alignments [56] and PhyloSuite v1.2.1 for concatenation [73].

In the ML inference, RAxML v8.2.12 was employed with 50 runs and 1000 bootstrap replicates using the GTRCAT model [74]. Convergence was then examined using the command “-I autoMRE”. After determining the best models in ModelTest-NG [75], MrBayes v3.2.7a was run to generate the Bayesian cladograms for 10,000,000 generations, with two runs using four chains each (sampling every 1000 generations). Furthermore, we tested the convergence of independent runs using Tracer v1.7.1 [76].

## 3. Results

### 3.1. Base Composition Comparisons of Plastid Genes among Saxifragales

To assess the impact of base compositions on codon bias in genes in the 361 individuals of the Saxifragales group, the GC content of 50 plastomic genes was counted individually. Overall, Appendix A shows extensive variation in GC content (GC123) among the genes, from a low of 28.91% in the *ycf1* gene of *Gonocarpus micranthus* from Haloragaceae to a high of 48.31% in the *rps11* gene in *Crassula volkensii* from Crassulaceae. Significantly, the GC and GC3 contents of all investigated genes were less than 50%, suggesting that AT nucleotides and AT-ending codons are often preferred within Saxifragales. Interestingly, in the first and second positions of codons, despite the fact that most genes showed an AT bias, some genes exhibited a GC bias. For instance, across 361 individuals, the following genes harbored GC1 > 50%: *atpB*, *clpP*, *petA*, *psaA*, *psbB*, *psbC*, *psbD*, *rbcL*, *rps11*, and *rps12*, while *rps11* and *rps7* showed GC2 > 50%.

To explore the interfamilial differences, the comparison of base compositions among the 11 families was performed. Our findings, summarized in Table 1 with mean and standard deviation, revealed that the patterns of GC content among different families were remarkably similar. Notably, the average GC values among three major clades of Saxifragales were highest in the Paeoniaceae + woody clade, with Paeoniaceae at the top with 39.46%, followed by Altingiaceae at 39.29%, Daphniphyllaceae at 39.20%, and Hamamelidaceae at 39.15%. Moreover, all 11 families exhibited a similar trend of GC-distribution across the three codon positions, with GC1 > GC2 > GC3.

### 3.2. Codon Usage Bias Indices of Plastid Genes among Saxifragales

The aim of this study was to examine the codon preference patterns among 361 Saxifragales individuals by calculating the ENC values of the 50 plastid genes (Appendix A). As shown in Table 2, the comparisons were initially performed at the intergenic level. The overall ENC values ranged from 39.23 ± 2.50 to 54.91 ± 3.38, with the lowest ENC found in the *rpl16* (39.23 ± 2.50) and the highest in the *pafI* gene (54.91 ± 3.38). The genes with lower ENC values, such as *rpl16*, *psbA*, and *rps8*, had stronger codon preferences, while those with higher ENC values, such as *pafI*, *ycf2*, and *rpl2*, showed weaker codon bias.

Then, to gain further insights into the interfamilial differences, the ENC values of 50 plastid genes were calculated as mean ± SD for each family and compared. The results displayed a high degree of diversity among 11 families (Appendix A and Figure 1). The most striking heterogeneity was identified in *ndhE* between Paeoniaceae (mean value: 58.44) and other ten families (mean values: 46.43–50.59), indicating a considerably lower bias in the former. Meanwhile, evidence from some taxa also supported the existence of a higher codon bias, particularly in the *rps8* genes of 26 Crassulaceae taxa and *Boykinia aconitifolia* (Saxifragaceae), with ENC < 35, suggesting significant codon usage preferences.

Furthermore, to explore intrafamilial diversities, we also compared the ENC values of genes within each family. Overall, a high level of variation was detected with the SD value ranging from 0 to 4.93. Remarkably, the most informative differences were detected in three families. For the *ndhE* genes in Haloragaceae, clade A and *Glischrocaryon aureum* had ENC values ranging from 43.20 and 45.57, while clade B had higher values of 48.78 to 50.49 (Figure 2a). Likewise, in the *cemA* genes of Hamamelidaceae, except for clade A with ENC of 50.88–53.68, clade B, C, D, E, and F similarly possessed lower a ENC range of 45.57–48.11 (Figure 2b). Strikingly, the most substantial intrafamilial difference was observed in the *atpF* genes of Altingiaceae (Figure 2c), where clade A could be clearly distinguished by its much higher ENC values of 51.42–52.07, compared to clade B and C, which had ENC values of 44.74–44.92.

The RSCU index was also used to assess codon usage bias. We performed the calculations on a concatenated matrix of 50 plastid genes from 11 families (Appendix A). In general, the plastid genes of Saxifragales harbored RSCU values ranging from 0.32 (CGC) to 2.16 (AGA). It was noted that 93.25% of the preferred codons with RSCU > 1 ended in AT, indicating an overwhelming bias for these codons in the CUB patterns. We further performed clustering analyses among the codons and families (Figure 3). Intriguingly, all families were clustered into the same branching patterns as those reported in the phylogeny trees of Saxifragales [18,19,20], with the exception of the positions of Iteaceae and Paeoniaceae.

### 3.3. Determinants in Codon Usage of Plastid Genes among Saxifragales

To understand the influences of mutation and selection on the CUB of plastid genes, PR2 bias plots were created focusing on the four-fold degenerate codons. Initially, the AT bias and GC bias at the third codon position were individually calculated in each gene (Appendix A). The ranges of AT bias (0.29 ± 0.02 to 0.67 ± 0.07) and GC bias (0.21 ± 0.09 to 0.84 ± 0.12) among the 50 genes of Saxifragales revealed a relatively high level of diversity. Then, our plot analyses also allowed the identification of unequal codon usage in different genes and families (Appendix A). Particularly, visibly family-specific variations were observed in two genes. The *petA* gene showed a stronger G bias in Crassulaceae compared to other families (Figure 4a). In addition, the *rps11* gene exhibited a higher T bias in Paeoniaceae than in other families (Figure 4b).

Significantly, several common patterns have also been discovered among the Saxifragales group. As illustrated in Figure 5, all 11 families collectively presented the CT bias in *atpA*, *psbA*, and *psbC*, the CA bias in *clpP* and *rps7*, as well as the GA bias in *rpl14*.

With the aim of inferring the main governing factors in the codon use patterns of plastid genes, ENC-GC3s plot and neutrality plot analyses were performed across Saxifragales. The results of the ENC plots revealed a similar pattern of point distribution among the 11 families (Figure 6). The GC3s values were relatively narrow (ranging from 0.14 to 0.35), and most of the splashes lay distant from the expected curve, suggesting that selection might play a bigger role in codon bias than mutation. Moreover, the gradient of regression line in the neutrality plot (GC12 vs. GC3) denoted the balance between mutation and selection (Figure 7). Across Saxifragales, the GC12 (0.45 ± 0.05) and GC3 (0.25 ± 0.04) were relatively concentrated, with a weak correlation (*r* = 0.02, *p* < 0.05), suggesting that mutation had a limited impact on the CUB. More importantly, the slopes of all 11 families were close to 0, with non-negative values ranging from 0.00593 (Altingiaceae) to 0.136 (Hamamelidaceae). The findings from both of the neutrality and ENC plots support the dominant role of selection pressure in shaping the CUB of plastid genes within Saxifragales.

### 3.4. Phylogenetic Inferences of Saxifragales Based on Plastid Data

We are dedicated to uncovering the phylogenetic relationships within the Saxifragales order. Our analysis utilized a 364-taxon matrix to generate a 77,368 bp alignment for further analyses. Based on two approaches, the ML and BI inferences generally resulted in well-resolved phylogenetic cladograms that shared similar topologies (Figure 8 and Appendix A).

Overall, our results suggested the existence of two main clades: Core Saxifragales (maximum likelihood bootstrap [BS] = 100 and Bayesian posterior probability [PP] = 1.00) and PWC (BS = 66, PP = 0.53). Two subclades of the former clade, the Crassulaceae alliance and the Saxifragaceae alliance, were both well recovered (BS = 100, PP = 1.00).

Within the Crassulaceae alliance, the Haloragaceae and Penthoraceae families are sister to the Crassulaceae, with robust supports (BS = 100, PP = 1.00). The families Saxifragaceae and Grossulariaceae (BS = 100, PP = 1.00), along with Iteaceae, were observed to be nested within the Saxifragaceae alliance. In addition, it is now recognized that the PWC clade consists of five families, with Hamamelidaceae + (Cercidiphyllaceae + Daphniphyllaceae) and Altingiaceae forming the sister group to Paeoniaceae. Notably, with the rare exceptions of Cercidiphyllaceae + Daphniphyllaceae (BS = 66, PP = 0.93) and the PWC node (BS = 66, PP = 0.53), all inter-family affinities received strong node support.

At a deeper phylogenetic level, we obtained high-resolution support (BS = 100, PP = 1.00) for the monophyly of all nine multiple-sample families. However, our analyses also revealed non-monophyly for some genera, including *Hylotelephium*, *Orostachys*, and *Sedum* in Crassulaceae, and *Distylium* in Hamamelidaceae.

In particular, our findings showed that *Pachyphytum compactum* was not assigned to its expected position based on the previous report, but rather clustered with *Echeveria lilacina* by weak support (BS = 55). Furthermore, at the deepest level, several individuals from the same taxon were not grouped together, but instead were nested within the clusters of other taxa, such as *Rhodiola quadrifida* (individual 2) and *Paeonia suffruticosa* (individual 5, 7, and 8).

## 4. Discussion

With the currently largest sampling of 361 Saxifragales plastomes, this study aimed to comprehensively evaluate the codon usage patterns of plastid genes in the Saxifrage order. Significantly, we employed many indices for analyses, including GC content at different positions of codons, RSCU, ENC, PR2 bias, ENC–GC3s, and neutrality plots. Most importantly, our findings presented here would considerably deepen our understanding of the evolutionary dynamics of plastid genes within the Saxifragales.

Interspecific codon usage heterogeneities are believed to be strongly influenced by GC content [77], which plays a vital role in the evolution of genome structure [34,78,79]. A high AT content has been widely recognized as an iconic characteristic of the plastome, which can be observed by a consistent pattern at the third codon position [44,80,81]. Similarly, all involved plastid genes in our investigation showed a tendency to utilize AT bases and AT-ending codons. It is noteworthy that a neutral effect of mutation on the third codon position leads to random usage of synonymous codons, resulting in a proportional usage of GC and AT [82,83]. Consequently, AT is strongly favored at the third bases of codons in the Saxifragales plastid genes, suggesting that selection pressure is responsible for the CUB patterns. Additionally, our findings revealed that ten genes in Saxifragales had GC1 > 0.5 and two genes had GC2 > 0.5, which is different from other studies where GC1, GC2, and GC3 values of plastid genes were all less than 0.5 [36,62,84,85]. Most notably, as proposed by Morton and So [86], stronger codon adaptation in plastid genes would lead to higher GC contents in the first and second bases of codons. Therefore, the 12 plastid genes mentioned above, particularly *rps11* with both GC1 and GC2 values over 0.5, might undergo higher codon adaptations during the evolution of the Saxifrage order. These findings will definitely provide a better understanding of the various evolutionary processes involving CUB across different plastid genes in the Saxifragales group.

The absolute synonymous codon bias can be accurately quantified by the ENC index, which measures the preferential degree of codon in a gene [19,44]. Importantly, it has been proposed that genes with higher expression tend to harbor lower ENC values and stronger CUB, and vice versa [60,87,88]. Moreover, Bulmer et al. [89] suggested that natural selection increases with the extent of gene expression. The plastid genes of Saxifragales exhibit a range of codon bias levels, both high and low. In particular, the highest bias was detected in *rps8* of 27 taxa (ENC < 35), which were mostly from Crassulaceae. Conversely, much weak bias was found in *ndhF* of Paeoniaceae with an average ENC > 55. Thereby, these findings not only highly mirrored the various expression levels of plastid genes among Saxifragales families, but also indicated that *rps8* might undergo more selection pressure in Crassulaceae. Additionally, the most substantial differences in ENC within each of the three families (Altingiaceae, Haloragaceae, and Hamamelidaceae) were found to harbor strong phylogenetic implications, which can further serve as clade-specific markers.

As another important index for CUB assessment, RSCU values could directly reflect the usage of synonymous codons. Thus far, it has been widely applied to explore codon use variations among genes. Across Saxifragales, the vast majority of the preferred codons were revealed to end with A/T bases. Notably, such bias might be congruent with its plastomic AT-rich base composition, and is also similar to the patterns of most dicot genomic genes [90,91,92]. In addition, the RSCU values of all NCG codons in Saxifragales families were less than 1, implying their relative rarity. Interestingly, it has been proposed that the infrequent usage of NCG codons might indicate a high level of methylation in the organism [93]. Our results demonstrated here could deepen our insights into the evolutionary patterns of plastid genes in the Saxifragales.

The silent (i.e., synonymous) sites of codons have been considered a powerful candidate to investigate evolutionary processes. To explain the preference for codon usage, several theoretical models have been proposed, comprising the neutral theory [94], the expression regulation theory [95], and the selection–mutation drift (SMD) theory [96].The SMD model, which attributes CUB to the combined consequence of selection, mutation, and drift [96,97,98,99], has received widespread attention. To date, numerous studies have supported its applicability in elucidating codon bias [42,100,101,102,103]. Of particular interest are the conclusions drawn for plastomic genes of angiosperms, where two different perspectives have emerged concerning the governing factors in CUB. Morton [104] argued that mutation and generic drift had a stronger influence, while another model suggests that selection plays a larger role, which has received support from multiple studies [105,106,107].

In this study of the Saxifragales group, the origin of the CUB in plastid genes appears to have been influenced by three factors, as suggested by the SMD model. One piece of evidence supporting this model is the presence of several splashes in the ENC–GC3s plots that are just on and near the curve, which suggests a partial role of mutation/drift [42,47]. However, the dominant role of selection is indicated by several factors, such as the obvious AT bias of the third codon position, the weak correlation between GC12 and GC3, the distant location of most splashes from the curve in the ENC plots, and the near zero slopes of neutrality plots. These findings collectively suggest that selection played a major role in shaping the CUB of Saxifragales plastid genes.

The rapid radiation in Saxifragales has resulted in highly divergent morphology, constituting a significant obstacle to disentangle its taxonomy [20]. Along with the great development of phylogenomic methodology, numerous efforts have been made to better understand its internal phylogenetic relationships [1,15,16,17,19,20,22]. Despite the placements of most constituent families having been well recovered to date, the relationships within the PWC clade are still a matter of debate [1].

Here, our phylogenetic inferences assigned 361 individuals into two main clades and four subclades. In general, the topology of the yielding trees was highly similar to those of the previous work [1,19,20]. Notably, most nodes received relatively high support, but some nodes had issues. For instance, we found that Paeoniaceae was a weak sister to the woody group, and a sister relationship existed between Cercidiphyllaceae and Daphniphyllaceae in the woody group, consistent with the findings of Han et al. [19] and Jian et al. [1]. However, these relationships were poorly resolved. In fact, the placement of Paeoniaceae has long been a taxonomic enigma, with inconsistent and weak support in previous analyses [1,7,15,16]. Additionally, according to the phylogeny analyses by multiple datasets of Jian et al. [1], the placement of Paeoniaceae resulted as an important inconformity among nuclear, mitochondrial, and plastid topologies of Saxifragales. Fishbein et al. [15] argued that the longest branch leading to this family could best explain such poor resolution, while Jian et al. [1] attributed the unrecovered relationships of the woody clade to an ancient, rapid radiation.

In addition, the unexpected phylogenetic positions of several taxa, as noted above, might be due to the following reasons: (1) species identification errors, (2) the current phylogenetic dataset’s inability to distinguish between extremely closely related taxa, and (3) potential chloroplast capture events. Therefore, acquiring more data and increasing sampling efforts will be crucial for gaining a deeper understanding of the phylogenetic affinities within Saxifragales.

## 5. Conclusions

This study aimed to present a comprehensive analysis of the codon utilization of the plastid genes, focusing on a larger-sampling scale of Saxifragales. Evaluations of CUB were performed with a series of indices at the inter- and intra-family levels, involving a total of 11 families. Overall, a variety of codon bias was elucidated among the 50 plastid genes. Despite a general bias toward AT bases and AT-ending codons, the GC1 and GC2 were still found to be more than 50% in 12 genes. Furthermore, the ENC values exhibited a high level of diversity among the constituent families. Notably, significant phylogenetic implications were identified for the intrafamilial heterogeneities of ENC in Altingiaceae, Haloragaceae, and Hamamelidaceae. Moreover, PR2 plots revealed that the *petA* genes in Crassulaceae and the *rps11* genes from Paeoniaceae were more G biased and T biased, respectively, compared to the other ten families. Furthermore, six genes exhibited favored base patterns that were shared by all the examined Saxifragales lineages. More importantly, both the ENC plots and neutrality plots suggested that selection pressure predominantly governed the CUB of Saxifragales plastid genes. Furthermore, the phylogenetic affinities inferred by both the ML and BI methods were consistent with each other, comprising two primary clades and four subclades. Altogether, our findings presented herein will convincingly deepen the current understanding of the molecular evolution of Saxifragales.

## Figures and Tables

**Figure 1 genes-14-00694-f001:**
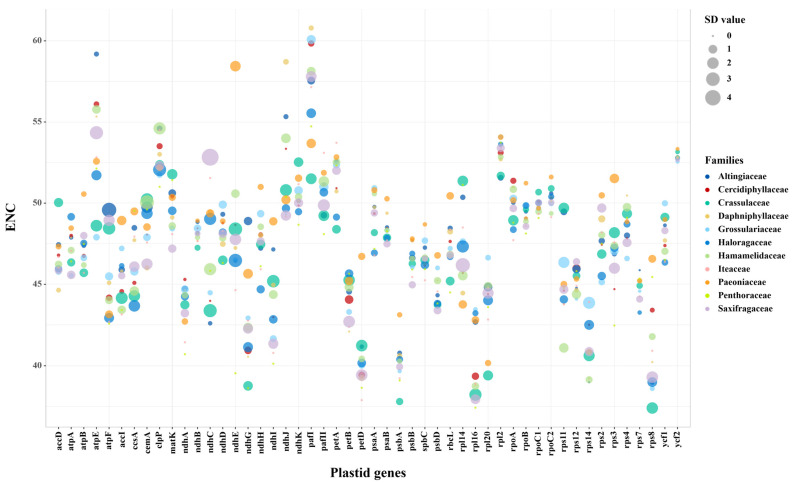
The mean ± SD values of ENC for each gene in each family of Saxifragales. The center, size, and color of circle denote the corresponding average value, SD value, and family, respectively.

**Figure 2 genes-14-00694-f002:**
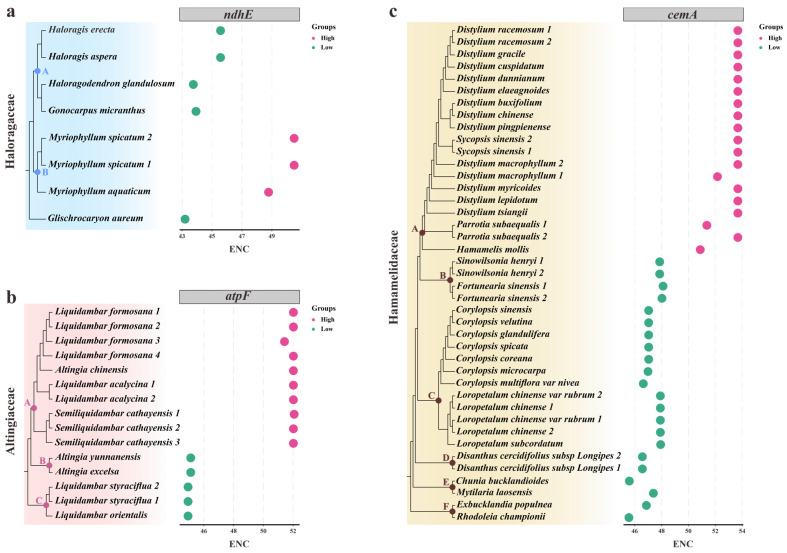
The different ENC patterns of plastid genes in three families. (**a**) The ENC values of *ndhE* genes in Haloragaceae taxa. (**b**) The ENC values of *atpF* genes in Altingiaceae taxa. (**c**) The ENC values of *cemA* genes in Hamamelidaceae taxa. Different clades are marked with dots in the nodes.

**Figure 3 genes-14-00694-f003:**
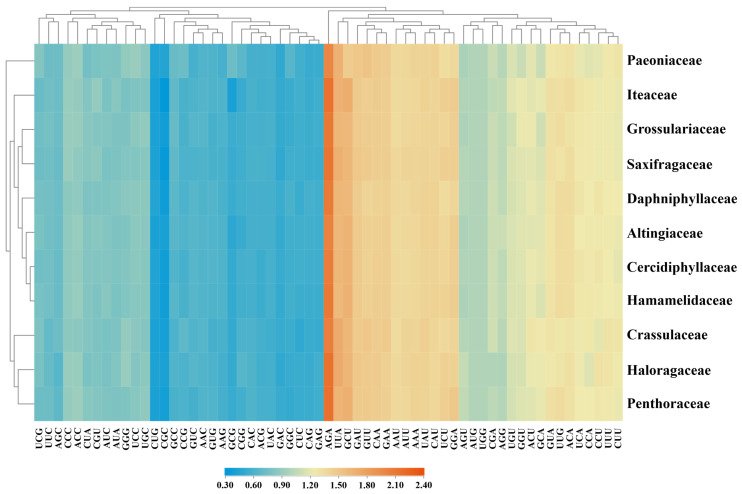
Heatmap of the RSCU values of the 50 plastid genes among Saxifragales. The horizontal axis represents the cluster of the RSCU values of codons, and the vertical axis represents the cluster of the constituent families.

**Figure 4 genes-14-00694-f004:**
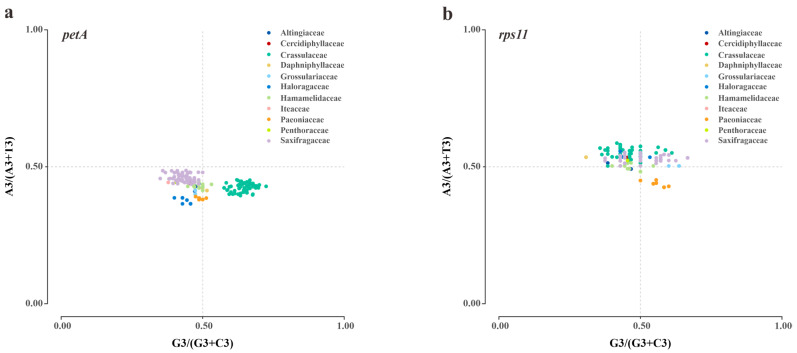
The PR2 bias plots of two plastid genes in Saxifragales. The colors denote the corresponding families. (**a**) *petA*; (**b**) *rps11*.

**Figure 5 genes-14-00694-f005:**
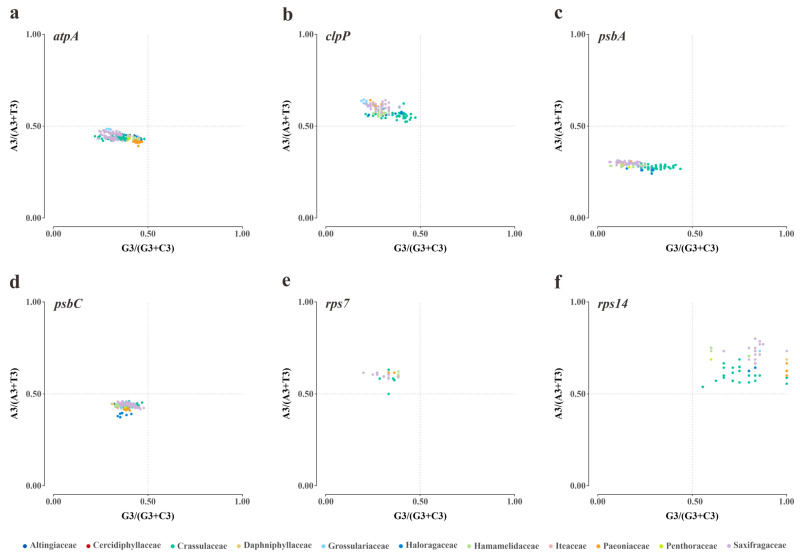
The PR2 bias plots of six plastid genes in Saxifragales. The colors denote the corresponding families. (**a**) *atpA*; (**b**) *clpP*; (**c**) *psbA*; (**d**) *psbC*; (**e**) *rps7*; (**f**) *rps14*.

**Figure 6 genes-14-00694-f006:**
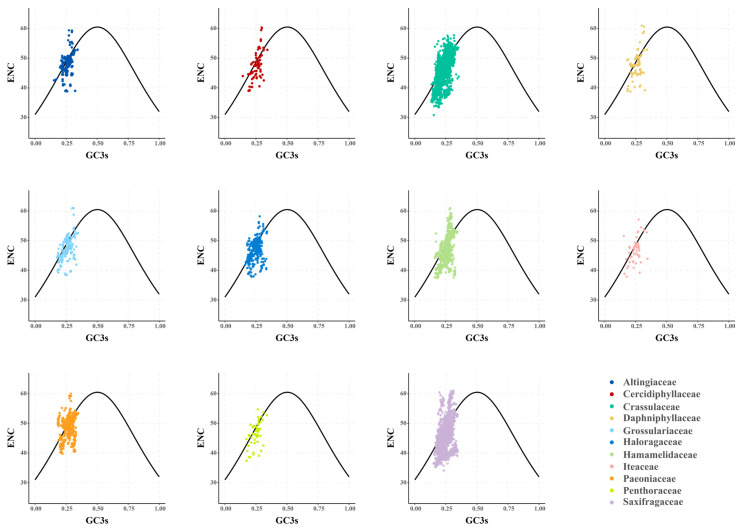
The ENC-GC3s plots of plastid genes in 11 Saxifragales families. The colors denote the corresponding families.

**Figure 7 genes-14-00694-f007:**
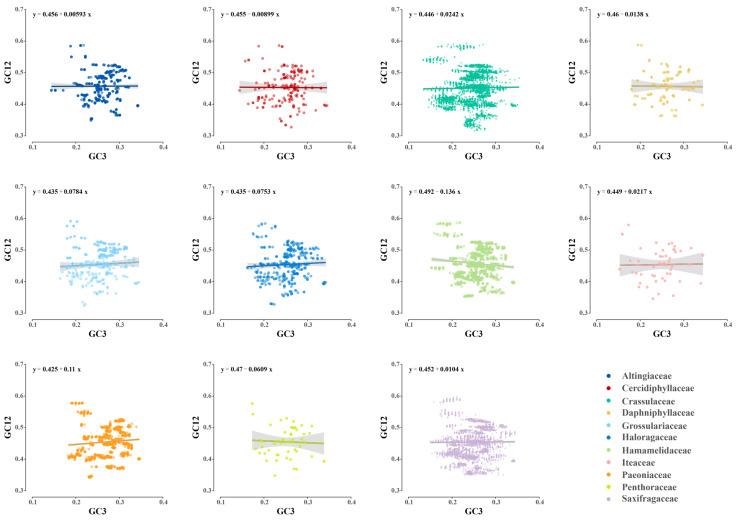
The neutrality plots of plastid genes in 11 Saxifragales families. The colors denote the corresponding families.

**Figure 8 genes-14-00694-f008:**
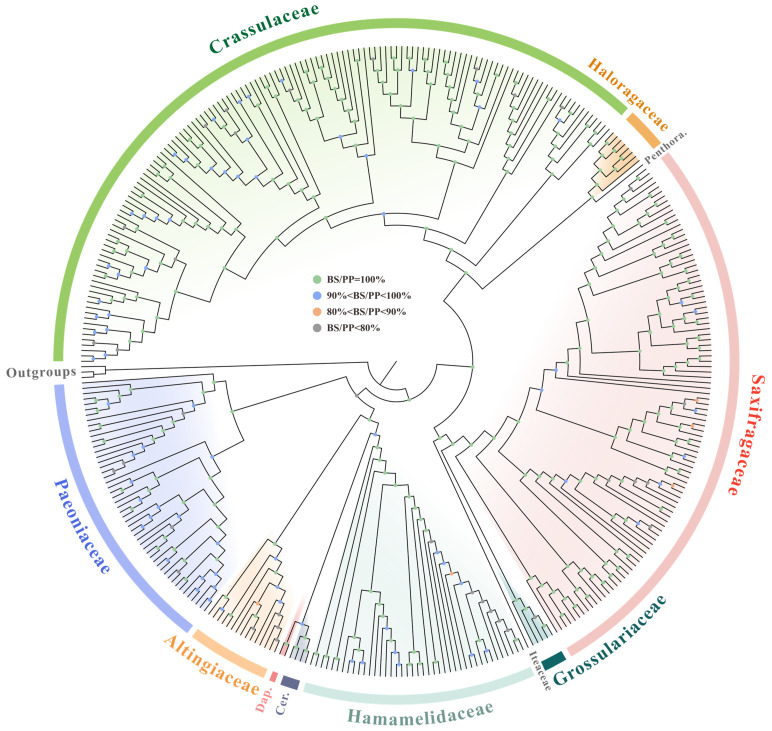
The simplified phylogenetic tree obtained by 79 PCGs among 361 Saxifragales plastomes, with BS and PP values shown by colored circles (the more detailed reconstruction is shown in Appendix A). Penthora. denotes Penthoraceae, Cer. denotes Cercidiphyllaceae, and Dap. denotes Daphniphyllaceae.

**Table 1 genes-14-00694-t001:** Base compositions of 50 plastid genes among 11 families of Saxifragales.

Family	GC%	GC1%	GC2%	GC3%
Altingiaceae	39.29 ± 3.31	48.47 ± 5.45	40.05 ± 5.75	29.35 ± 3.21
Cercidiphyllaceae	38.67 ± 3.44	48.05 ± 5.48	39.65 ± 5.83	28.31 ± 3.59
Crassulaceae	38.42 ± 3.53	47.84 ± 5.56	39.54 ± 5.76	27.89 ± 3.69
Daphniphyllaceae	39.20 ± 3.25	48.43 ± 5.38	40.01 ± 5.57	29.15 ± 3.53
Grossulariaceae	38.83 ± 3.44	48.18 ± 5.52	39.82 ± 5.58	28.49 ± 3.74
Haloragaceae	38.79 ± 3.45	48.15 ± 5.51	39.78 ± 5.61	28.45 ± 3.57
Hamamelidaceae	39.15 ± 3.19	48.36 ± 5.51	40.11 ± 5.64	28.97 ± 3.34
Iteaceae	38.59 ± 3.59	48.04 ± 5.76	39.87 ± 5.83	27.87 ± 3.96
Paeoniaceae	39.46 ± 3.47	48.26 ± 5.34	39.86 ± 5.55	30.27 ± 3.60
Penthoraceae	38.75 ± 3.30	48.26 ± 5.56	39.88 ± 5.51	28.12 ± 3.57
Saxifragaceae	38.77 ± 3.43	48.23 ± 5.44	39.76 ± 5.77	28.33 ± 3.60

**Table 2 genes-14-00694-t002:** The mean ± SD ENC values of 50 plastid genes of Saxifragales.

Gene	ENC Values	Gene	ENC Values	Gene	ENC Values	Gene	ENC Values	Gene	ENC Values
*rpl16*	39.23 ± 2.50	*ndhI*	44.48 ± 3.03	*atpF*	47.13 ± 3.09	*rps2*	48.28 ± 1.81	*rpoC1*	50.16 ± 0.63
*psbA*	39.67 ± 1.82	*rps7*	44.77 ± 0.67	*atpB*	47.21 ± 1.78	*ycf1*	48.53 ± 1.09	*rpoC2*	50.54 ± 0.77
*rps8*	39.92 ± 3.46	*atpI*	45.28 ± 2.24	*ndhC*	47.27 ± 5.15	*rps4*	48.72 ± 1.40	*petA*	50.78 ± 2.10
*rps14*	40.60 ± 1.56	*rps12*	45.56 ± 0.95	*ndhD*	47.66 ± 1.36	*cemA*	48.76 ± 2.68	*ndhJ*	50.86 ± 2.45
*petD*	41.32 ± 2.81	*ccsA*	45.77 ± 2.42	*accD*	47.74 ± 2.03	*psaA*	49.24 ± 1.19	*ndhK*	51.21 ± 1.53
*ndhG*	41.66 ± 3.06	*psbB*	46.21 ± 1.06	*rps3*	47.80 ± 2.36	*rpoB*	49.45 ± 0.93	*atpE*	52.19 ± 3.73
*rpl20*	42.05 ± 2.88	*rps11*	46.40 ± 3.25	*ndhB*	47.92 ± 0.67	*rpoA*	49.62 ± 1.27	*clpP*	52.68 ± 1.53
*ndhA*	43.59 ± 1.06	*atpA*	46.66 ± 1.27	*rpl14*	47.94 ± 3.61	*ndhE*	49.68 ± 4.29	*rpl2*	52.72 ± 1.15
*psbD*	44.30 ± 1.33	*rbcL*	46.80 ± 1.89	*ndhH*	48.09 ± 1.57	*matK*	49.78 ± 2.19	*ycf2*	52.98 ± 0.28
*petB*	44.42 ± 2.04	*psbC*	46.87 ± 1.06	*psaB*	48.20 ± 1.09	*pafII*	50.11 ± 1.95	*pafI*	54.91 ± 3.38

## Data Availability

Not applicable.

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
