# Peer review of "Codon Usage Analyses Reveal the Evolutionary Patterns among Plastid Genes of Saxifragales at a Larger-Sampling Scale"

_genes, 2023, doi:10.3390/genes14030694_

Round 1

Reviewer 1 Report

The manuscript is interesting and well written.

Howevere, a few changes/corrections are needed.

The introduction section is a bit short and lacks the study's evolutionary background. 

Please clarify the objectives of the study.

Authors should mention the recession numbers of the sequences used the analysis and the link of the database from which they got sequence data.

Imrove the resolution of the figures 1,2 and 3.

Please carefully check the citations and bibliography throughout the manuscript.

Reviewer 2 Report

Peer review of Codon usage analyses reveal the evolutionary patterns among plastid genes of Saxifragales at a larger-sampling scale by Bi et al. for Genes

The authors perform a comprehensive examination of codon usage bias in the Saxifragales. Performing analyses like these are very important because we often use synonymous substitutions as something of a proxy for neutral evolution even though it seems that there is some degree of selection on synonymous sites; analyses like these have broad implications for studies of evolution that include analyses of selection that use synonymous sites like dn/ds, pn/ps, ka/ks, McDonald-Kreitman tests, etc.

The authors were thorough in their analyses and despite not finding any particularly surprising results, it is important that analyses like this are performed for the aforementioned reasons. This paper could, however, use a meaningful degree of proofreading and copyediting.

I have two comment on how the analyses need to be reported and discussed:

Line 137 – How was the GTRCAT model selected? Include the name of the model testing program and/or whatever details were used to select the model.

Line 254 – There is not as substantial sampling of the taxa or number of sites with nuclear markers but some discussion of Saxifragales phylogenies using nuclear markers and the degree to which these plastid-based analyses agree with them would be warranted. There are references to some studies that use some nuclear markers but this issue (the degree of concordance between nuclear-based analyses and your plastid-based analyses) should be discussed some.

I have not done a substantial proofreading or editing, which I think this manuscript needs, but I have noticed two grammatical issues that I spotted and want to make sure are addressed:

Line 56 – “codons” should be “codon”

Line 347 – “to have influence” should be “to have been influenced by”

I am recommending this manuscript for minor revisions. I look forward to seeing  the addressing of the minor issues I raised above and a thorough process of proofreading and copyediting.

Reviewer 3 Report

The authors have provided a comprehensive assessment of the codon manipulation based on the 50 plastid genes, covering 11 families from Saxifragales. The paper is well written, and results are interpreted scientifically. There are a few concerns which authors need to address:

1. The authors have mentioned that the Saxifragales constitutes 15 families, but yet authors have only taken 11 families for their study. If possible, then authors should include all the 15 families in their study so that it would strengthen and would make this work more robust.

2. The authors should also justify as to why they have chosen the 50 plastid genes to assess the codon manipulation. Is it possible to do the same study with other set of genes and would it give similar results?

Round 2

Reviewer 3 Report

The concerns raised are fairly explained by the authors